# Combination of Adult Mesenchymal Stem Cell Therapy and Immunomodulation with Dimethyl Fumarate Following Spinal Cord Ventral Root Repair

**DOI:** 10.3390/biology13110953

**Published:** 2024-11-20

**Authors:** Paula Regina Gelinski Kempe, Mateus Vidigal de Castro, Lilian de Oliveira Coser, Luciana Politti Cartarozzi, Benedito Barraviera, Rui Seabra Ferreira, Alexandre Leite Rodrigues de Oliveira

**Affiliations:** 1Laboratory of Nerve Regeneration, University of Campinas (UNICAMP), Campinas 13083-865, SP, Brazil; paulakempe@gmail.com (P.R.G.K.); mateusvidigal@hotmail.com (M.V.d.C.); coser.lilian@gmail.com (L.d.O.C.); lpcarta@unicamp.br (L.P.C.); 2Center for the Study of Venoms and Venomous Animals (CEVAP), São Paulo State University (UNESP), Botucatu 01419-901, SP, Brazil; benedito.barraviera@unesp.br (B.B.); rui.seabra@unesp.br (R.S.F.J.); 3Center for Translational Sciences and Biopharmaceuticals Development, Botucatu 18610-307, SP, Brazil; 4Medical School, São Paulo State University (UNESP), Botucatu 18618-687, SP, Brazil

**Keywords:** spinal cord injury, stem cell therapy, gliosis, inflammation

## Abstract

Spinal cord injury affects millions of individuals worldwide, underscoring the urgent need for more effective treatments that can facilitate the recovery of motor and sensory functions. The use of neuroprotective molecules in conjunction with stem cell therapy may facilitate the restoration of function, thereby enabling patients to resume a productive lifestyle. The present study demonstrates that the combination of dimethyl fumarate, an immunomodulatory drug, with adult mesenchymal stem cells derived from adipose tissue exhibits neuroprotective and immunomodulatory properties. Furthermore, we observed a notable rescue of spinal motoneurons following the avulsion and reimplantation of ventral roots. Additionally, glial reactions were reduced, which may contribute to enhanced long-term outcomes. The collective findings support the potential of integrated regenerative strategies following spinal cord root injury.

## 1. Introduction

The motor root avulsion injury model is a widely used methodology for investigating neuronal death pathways and potential neuroprotective therapies. After injury, motoneurons degenerate through necrosis and apoptosis events [1,2]. Motoneurons that degenerate due to necrotic events present biomembrane fragmentation and extravasation of cytoplasmic content into the extracellular environment, which, in turn, leads to macrophage activation and complement system activation, triggering the initial inflammatory events in the injured tissue. Subsequently, injured motoneurons degenerate as a consequence of internal signaling that activates apoptotic pathways [3,4,5,6,7]. This may occur if the neuron exhibits low regenerative capacity or an inability to regenerate or maintain connectivity with appropriate networks [1,2,8,9]. Thus, it is possible to observe DNA fragmentation and chromatolysis, atrophy of the nucleus and its displacement to the periphery, and a reduction in the size of the cell body and dendrite complexity [1,8]. Furthermore, the inflammatory response is initiated shortly after the injury progresses, with the activation of astrocytes and the recruitment of macrophages to the injured site. This results in an intense inflammatory response with the production of pro-inflammatory cytokines [9,10]. Neurons with high regenerative capacity can bypass these events through the upregulation of survival-related genes and neurotrophic factors that modulate the microenvironment in an attempt to survive [1,11,12].

Surgical repair, achieved through suture [13], grafting, or the use of biological glues or biopolymers [14,15], represents a crucial strategy for motor unit recovery in the motor root avulsion model. The heterologous fibrin biopolymer (HFB), derived from snake venom, acts as a biological glue and is a more advantageous treatment than suturing or grafting. It minimizes damage to healthy tissue, prevents further harm to injured tissue, and forms a biological mesh, allowing treatment with added stem cells [15]. HFB has been shown to possess neuroprotective capabilities, promoting the recovery of synaptic circuits, controlling tissue inflammation, inducing trophic factor production, and fostering axonal regeneration. These beneficial effects have been linked to improvements in sensory and motor function, as evidenced by several studies [9,14,15,16,17,18].

Dimethyl fumarate (DMF) has proven to be a promising treatment option for multiple sclerosis due to its neuroprotective and immunomodulatory properties [19,20]. It has been employed in a number of CNS disease models demonstrating effects such as neuronal preservation, maintenance of synaptic circuitry, decreased inflammatory reactions, regeneration of nerve fibers, and improved motor function [14,21,22,23,24,25]. The beneficial effects of DMF are primarily attributed to the upregulation of the NRF2 factor, which is associated with the transcription of antioxidant enzymes, including glutathione-S-transferase-A2 (GSTA2), heme oxygenase-1 (HO1), and NADPH-quinone oxidoreductase-1 (NQO1) [25,26].

Moreover, DMF exerts influence over immune cells, which are involved in the inflammatory response, and which are crucial for nerve myelination [19,27,28,29]. Its impact on Schwann cells is evident in the preservation and regeneration of myelinated fibers, further contributing to improved outcomes [18,30]. Furthermore, DMF has also been demonstrated to exert beneficial effects on glial cells following motor root avulsion injury, particularly astrocytes and microglia. These cells are responsible for the inflammatory activation of tissue and the clearance of debris resulting from neuronal death [14]. Additionally, astrocytes play a role in the retraction and elimination of synaptic terminals following injury [31,32,33]. This is consistent with the observation that DMF is capable of preserving synaptic coverage, as it has been shown to reduce astrocytic activity [18].

Recent studies have indicated that mesenchymal stem cell (MSC) therapy may have promising effects in the treatment of spinal cord injury and neurodegenerative diseases [9,10,34,35]. Given their capacity to produce chemokines, interleukins, and trophic factors such as NGF, BDNF, and GDNF [36,37,38], MSCs demonstrate considerable potential for neuroprotective and anti-inflammatory effects. It has been demonstrated that the transplantation of MSCs is an efficacious strategy in the acute injury phase, which corresponds to the initiation of apoptotic and inflammatory processes [39,40,41,42].

Consequently, we hypothesized that the combination of pharmacological and stem cell therapy approaches can enhance the rescue of avulsed motoneurons as well modulate inflammation, preserve the synaptic circuitry in the motor nucleus (lamina IX of Rexed), and facilitate motor function recovery after ventral root avulsion reimplantation.

## 2. Materials and Methods

### 2.1. Study Experimental Design

The objective of this study was to investigate the efficiency of a multi-therapeutic approach for the treatment of experimental ventral (motor) root avulsion in a rat model. The treatments employed included the use of a HFB for reimplantation of the avulsed roots at the site of injury, in combination with administration of MSC therapy and DMF. A total of 115 eight-week-old young adult female Lewis rats (LEW/HsdUnib) were used in this study. Such a choice was based on the previous literature and on the fact that young adult rats are more suitable to long-term spinal cord injury and repair approaches [10,14,18,39,43]. Female rats were used due to the long-term limited size and weight gain, which facilitate handling and functional tests such as the walking track test. The animals were provided with ad libitum access to food and water in a 12 h light/dark cycle. All animals underwent surgical avulsion of the L4, L5, and L6 ventral roots (VRA) on the ipsilateral side. Following avulsion, the motor roots were either directly reimplanted with HFB or left untreated. Some animals were treated with MSC treatment while others received DMF or a combination of both.

The study is comprised of three main experiments. In Experiment I, the objective was to compare two types of easily obtained MSC derived from adipose tissue (AT) and bone marrow (BM), with a particular focus on neuronal survival and synaptic preservation. In Experiment II, the AT-MSC lineage, which had demonstrated superior results in terms of synaptic preservation, was combined with pharmacological treatment using DMF and lesion repair with HFB. In this regard, we evaluated the combined impact of therapies on gait recovery, neuroprotection, synapse preservation, and tissue inflammation over a 12-week period. Finally, in Experiment III, we assessed the combined effect of therapies on gene expression during the acute post-injury period (one week after VRA). Each experimental group consisted of N = 5 or 6 animals. Table 1 shows the experiments carried out as well as the respective experimental groups.

The HFB is comprised of three components, which are homogenized and administered in sequence at the lesion site, with a final volume of 6 µL. The first fraction is a fibrinogen cryoprecipitate (3 μL), which is derived from the blood of *Bubalus bubalis*. The second fraction is the calcium chloride diluent (2 μL). The last fraction, the gyroxin (1 μL), is a thrombin-like enzyme from *Crotalus durissus terrificus* venom and is responsible for accelerating the biopolymer formation [44]. The HFB components and associated formulas are detailed within the patent BR1020140114327, and were generously provided by the Center for the Study of Venoms and Venomous Animals (CEVAP, UNESP, Brazil).

The administration of DMF (Sigma-Aldrich, St. Louis, MO, USA, Cat. number 242926) was initiated 30 min after surgery. DMF, diluted in a 0.08% methylcellulose solution, was administered orally by gavage at a dose of 15 mg/kg daily for 4 weeks [14]. It is important to note that, although one of the survival times is 12 weeks, we only administered DMF up to the fourth week. This decision was made due to the potential adverse effects of the drug, with the goal of a shorter and more controlled treatment, based on a previous article [14]. The 0.08% methylcellulose solution was used as the vehicle. The contralateral side of the spinal cord was used as an internal control for all analyses.

### 2.2. MSC Isolation and Culture

BM-MSCs and AT-MSCs were harvested from eGFP transgenic female Lewis rats at 8–10 weeks of age. The animals were euthanized with isoflurane. For the extraction of BM-MSCs, tibiae and femurs were collected, and bone marrow was removed using a 20 mL syringe and a 16-gauge needle. The collected bone marrow was centrifuged (1200 rpm, 3 min) and the precipitate was resuspended in DMEM basal culture medium with 10% FBS. The resuspended material was plated into a T-25 cell culture flask [36].

For extraction of AT-MSCs, approximately 2 g of inguinal adipose tissue was harvested, fragmented with scissors, and treated with DPBS and 0.2% collagenase type I. Fragments were incubated at 37 °C for 60 min with vigorous agitation for 1 min every 15 min. After filtration through a 40 µm porosity cell strainer, collagenase was inactivated with DMEM supplemented with 10% FBS. After two 10 min centrifugations at 1500 rpm, the precipitate was resuspended and seeded in DMEM supplemented with 10% FBS and seeded in a T-25 cell culture flask [45].

Both MSC lines were incubated at 37 °C with a 5% CO_2_ atmosphere. After two days, non-adherent cells and debris were removed by washing with PBS. A homogeneous monolayer culture was maintained by successive medium changes over two weeks. Routine passaging by trypsinization was performed at 80–90% confluence.

### 2.3. MSC Characterization

Cells were incubated with anti-CD90-PE-Cy7, anti-CD54-PerCP, anti-CD45-PE, and anti-RT1A-PE antibodies (BD Bioscience, San Diego, CA, USA) (Appendix A) for 30 min at 4 °C. After incubation, the cells were centrifuged at 400× *g*, 5 min, 4 °C, and washed with PBS-BSA-A. Cells were then fixed with 150 µL of fixation buffer (True Nuclear Transcription Buffer Set; Biolegend, San Diego, CA, USA), according to the manufacturer’s instructions.

Cells were analyzed using a NovoCyte Flow Cytometer (ACEA Biosciences, San Diego, CA, USA) and the data were analyzed using NovoExpress software (version 1.6.2); unlabeled cells were used as a negative control.

### 2.4. VRA and Lesioned Motor Root Reimplantation

The animals were intraperitoneally anesthetized with a combination of xylazine hydrochloride (Anasedan^®^, Ceva Santé Animale, Paulínia-SP/Brazil 10 mg/kg) and ketamine hydrochloride (Dopalen^®^, Ceva Santé Animale, Paulínia-SP/Brazil.50 mg/kg) at a 1:1.5 ratio. Once the reflexes had been lost, deep anesthesia was maintained with 2% isoflurane. To ensure ocular comfort, a lubricating ophthalmic gel (Liposic, Bausch & Lomb, Porto Alegre, Brazil 2 mg/g; 0.2% *w*/*w*) was applied, and the animals were placed on a heated pad throughout the procedure. In a prone position, the ventral lumbar roots were exposed via laminectomy of approximately two vertebrae (T13-L1). A unilateral avulsion of the ventral lumbar roots (L4, L5, and L6 segments) was conducted. The roots were promptly repositioned to their original sites and reimplanted with HFB in a total volume of 6 μL. This consisted of fibrinogen derived from buffalo blood (3 μL), calcium chloride (2 μL), and thrombin-enriched fraction (1 μL) [14]. The stability of root fixation was a determining factor in the success of the repair. Post-surgical recovery occurred in a heated environment for four hours. Tramadol hydrochloride (Germed Farmacêutica Ltd Campinas, Brazil was administered at a dose of 0.5 mg/kg by gavage every 24 h until the third postoperative day.

### 2.5. MSC Transplantation

The MSCs were gently engrafted at the interface between the CNS and the PNS, immediately following the avulsion and motor root repair. A total of 3 × 10^5^ cells were diluted in 5 μL of DMEM and applied over the HFB scaffold at the injury site. DMEM served as a control for cell therapy.

### 2.6. Specimen Preparation

At one, four, and twelve weeks post-lesion, the animals were euthanized with a lethal intraperitoneal dose of a xylazine hydrochloride (30 mg/kg) and ketamine hydrochloride (150 mg/kg) combination. Once the animals had lost all reflexes, they were perfused transcardially with phosphate buffered saline (PBS).

For qRT-PCR analysis, lumbar intumescence was collected, divided into ipsi and contralateral blocks, rapidly frozen in liquid nitrogen, and stored at −80 °C until use. Additionally, animals were perfused with 4% paraformaldehyde in PB 0.1 M (pH 7.4) for Nissl staining and immunofluorescence analysis. The lumbar intumescence was subsequently collected and postfixed for 24 h at 4 °C in the same fixative solution. The samples were subjected to a series of sucrose solutions (10%, 20%, and 30%) for 24 h each at 4 °C. The samples were embedded in Tissue-Tek (Miles Inc., Santee, CA, USA) and frozen at −35 to −40 °C. A Cryostat (Micron HM25) was used to obtain 12-μm-thick cross-sections of the spinal cord, which were subsequently stored at −20 °C until use.

### 2.7. Neuronal Survival Assessment

The number of motoneurons were determined by the counting of cells in cross-sections of the lumbar intumescence. This was performed using the Nissl-stained slides (0.05 g of toluidine blue in 1000 mL of distilled H_2_O). For that, spinal cord cross-sections underwent a 30 min acclimatization, followed by 30 s staining. After washing in distilled water, the slides underwent a series of alcohol dehydration steps (70%, 80%, 90%, and 100%), concluding with immersion in xylene. Subsequently, the slides were mounted using Entellan (Sigma-Aldrich) and a coverslip.

### 2.8. Immunofluorescence

The slides with spinal cord cross-sections were thawed at room temperature, washed with phosphate buffer (PB 0.01 M) for 3 × 5 min, and incubated with a blocking solution (3% BSA in PB 0.1 M) for 45 min. Subsequently, the slides were incubated for 3 h with primary antibodies (see Appendix A) in an incubation solution (1% BSA + 0.2% Triton in PB 0.1 M) at room temperature. Afterward, the slides were washed with PB 0.01 M (3 × 5 min), and the respective secondary antibodies (Alexa 488 or 594) were incubated for 45 min. The slides were again washed in PB 0.01 M (3 × 5 min) and mounted in glycerol (66% of glycerin in PB 0.1M) with DAPI (1:1000).

The specimens were observed under a Leica DM5500 microscope, and three representative images were captured (Leica DFC-345FX) from each sample for each animal on both ipsi and contralateral sides of lamina IX of Rexed at the ventral column of the spinal cord at 20× or 40× magnification.

The integrated density of pixels, representing immunostaining intensity, was quantified using ImageJ software (version 1.51 h), and the entire image was used for analysis. The results are expressed as the ipsilateral/contralateral ratio for each animal.

### 2.9. RT-qPCR

To obtain total RNA from spinal cord samples, QIAzol (QIAGEN, 79306) was added and subjected to Polytron lysis and homogenization. The resulting solution was then incubated for 5 min at room temperature to dissolve core-protein complexes. Subsequently, chloroform was added, followed by vigorous stirring. The samples were incubated for 5 min at room temperature, followed by centrifugation (12,000 rpm, 15 min, 4 °C). The upper aqueous phase, containing the RNA, was collected and mixed with 500 µL of isopropyl alcohol followed by incubation for 10 min and centrifugation (12,000 rpm, 10 min, 4 °C). The supernatant was carefully aspirated and 1 mL of 75% ethyl alcohol was added, followed by centrifugation (7500 rpm, 5 min, 4 °C). The supernatant was completely removed, and the RNA pellet was air-dried for 5 to 10 min. Finally, the RNA was eluted in 20 μL of RNAse-free water and incubated for 10 min in a dry bath at 55 °C.

Following confirmation of the quantity and quality of RNA, complementary DNA (cDNA) synthesis was performed using the High-Capacity cDNA Reverse Transcription Kit (Applied Biosystems, 4368814). cDNA was synthesized from 1.5 μg of total RNA and was utilized for further steps. The RT-qPCR reactions were performed in triplicate, including: the cDNA, TaqMan^®^ Gene Expression Master Mix (Life Technologies-PN 4369016), RNAse-free water, and Taqman assays (containing primer + hydrolysis probes) for the genes contained in Appendix A, making a final volume of 10 μL. HPRT1 was used as a reference gene.

The entire RT-qPCR procedure was conducted on the MX3005P instrumentation platform, with thermocycling as indicated by the Master Mix: 1 cycle of 10 min at 95 °C followed by 45 cycles of 95 °C for 15 s and 60 °C for 1 min. MxPro software Version 4.10 presented the results, and the relative quantification of the genes of interest was determined by the 2^−ΔΔCt^ method [46].

### 2.10. Motor Function Recovery Assessment

The CatWalk system (CatWalk XT 5.0, Noldus) comprises an illuminated platform that enables the data acquisition, including paw print size, pressure, width, and length. These parameters provide insights into several key aspects of the animal’s behavior, including strength (maximum contact and paw intensity) and motor coordination (peroneal nerve function and paw positioning).

Two measurements were obtained as baseline values for each animal prior to injury. Post-injury assessments were conducted at 3-day intervals over a 12-week period, with four recording sessions per animal at each evaluation. The mean values were calculated for each baseline measurement and on each day of analysis for each animal. The mean and standard error of the mean (SEM) for each experimental group were calculated.

### 2.11. Statistical Analysis

The mean and SEM were calculated for the numerical results within each group. The following statistical methods were employed for specific datasets: one-way ANOVA followed by Tukey’s posttest for neuronal survival; immunofluorescence and RT-qPCR; two-way ANOVA; and Mann–Whitney for functional analysis data (walking track test). Significance levels were assumed when * *p* < 0.05, ** *p* < 0.01, *** *p* < 0.001, and **** *p* < 0.0001 for all analyses. All statistical analyses were conducted using Prism 6 software.

## 3. Results

### 3.1. Characterization of MSC Cultures

The evaluation of 50,000 events (cells) by flow cytometry revealed that the percentage of cells expressing CD90 was 99.98% for AT-MSCs and 99.89% for BM-MSCs; the percentage of cells expressing CD54 was 99.48% for AT-MSCs and 92.01% for BM-MSCs; and the percentage of cells expressing CD45 was 0.53% for AT-MSCs, and 7.71% for BM-MSCs (indicating low hematopoietic cell presence). RT1A exhibited a percentage of 5.86% for AT-MSCs and 2.43% for BM-MSCs, suggesting a low occurrence of MHC class I (Appendix A).

Furthermore, positive immunostaining for CD90 and negative immunostaining for CD45 were observed, combined with DAPI staining. These findings confirm the specific surface marker expression and characteristics of mesenchymal stem cells, supporting their identity and suitability for the study. The engrafted AT-MSCs were detected in situ by immunolabeling for CD90 4 weeks after ventral root reimplantation with fibrin biopolymer. No co-localization with GFAP, Iba-1, or neurofilament was found, confirming the maintenance of the undifferentiated state over time (Appendix A).

### 3.2. AT-MSC Outperforms BM-MSC in Synaptic Preservation

A comparative analysis of AT-MSC and BM-MSC indicates that AT-MSC exhibits superior efficacy in maintaining neuronal survival and preserving synaptic integrity (Figure 1 and Appendix A). This is in line with AT-MSC anti-inflammatory capabilities, and is potentially achieved through the regulation of astrocytic reactivity and microglia activation.

In the absence of cell therapy or pharmacological treatment (vehicle-DMEM), a significant degeneration of injured motoneurons is observed after four weeks. It is noteworthy that both the AT-MSC and BM-MSC therapy groups exhibited preserved neurons in the motor nuclei (56% and 50%, respectively), with no statistically significant differences between them, which highlights their potential neuroprotective effects.

With regard to tissue inflammation, both cell therapy groups demonstrated a 50% reduction in microglia reaction and a 25% reduction in astrogliosis in comparison to the vehicle group. No statistically significant differences were observed between the two cell therapy groups. These anti-inflammatory effects are particularly relevant around avulsed neurons.

Concerning the overall synaptic preservation, a significant decrease (50%) is observed after injury. While the BM-MSCs demonstrated a modest capacity to preserve general synapses (62%), with no statistical differences compared to the vehicle group, AT-MSC treatment significantly preserved general synapses in the lesion microenvironment (78%).

These findings suggest that both MSC strains are neuroprotective and modulate the inflammatory response after injury. Therefore, AT-MSCs exhibit superior synaptic preservation, which places them as the preferred choice for subsequent experiments for the study of reimplantation with HFB and cell therapy in combination with DMF treatment.

### 3.3. Enhanced Neuronal Survival with Combined Therapies

The neuroprotective effect of the combined therapies was assessed by evaluating neuronal survival, expressed as the ipsilateral to contralateral ratio of motoneurons at 12-weeks post injury (Figure 2). Significant motoneuron degeneration was found in the VRR group, with 44% of neurons being preserved. In contrast, both groups treated with AT-MSCs showed improved outcomes, demonstrating 57% preservation in the VRR + AT-MSC group and 63% in the group with all therapeutic approaches (VRR + AT-MSC + DMF). Interestingly, no statistically significant differences were observed between the two cell therapy groups, demonstrating their comparable and substantial neuroprotective effects.

### 3.4. Decreased Glial Activity–Astrogliosis and Microgliosis

To evaluate glial reactivity 12 weeks after avulsion and ventral root reimplantation, anti-GFAP antibody assessed reactive astrogliosis and anti-Iba-1 antibody assessed microglial reaction in the lamina IX of Rexed (Figure 2, and Appendix A). The control group (VRR) exhibited twice as much glial reactivity, especially around the avulsed neurons. The VRR + AT-MSC group showed a reduction in microglia reaction, and the VRR + AT-MSC + DMF group exhibited an even more pronounced reduction. In terms of astrocytic activity, both cell therapy groups showed a significant reduction compared to the VRR group.

### 3.5. Inhibitory, Excitatory, and Overall Synaptic Preservation

Twelve weeks after avulsion and reimplantation, a reduction in GABAergic immunostaining was observed, indicating a loss of inhibitory synapses. The VRR group showed greater preservation of these synapses than the groups that received additional therapeutic approaches, albeit with a small difference of 6%. Conversely, analysis of excitatory glutamatergic inputs revealed an inverse pattern. The VRR group showed less preservation, whereas the groups that received cell therapy and DMF showed greater preservation, with a significant difference of approximately 20% compared to the VRR. This suggests that cell therapy and DMF treatment contribute to the maintenance of excitatory synaptic inputs.

When analyzing the overall synaptic preservation, regardless of the neurotransmitter content, we observed a synaptic density of 77% after reimplantation and cell therapy. In the VRR + AT-MSC + DMF group there was a substantial synaptic preservation in the dorsolateral motor nuclei, reaching 91% (Figure 3 and Appendix A).

### 3.6. Gait Recovery

Significant differences were observed in the hind paw print length and width interaction, as indicated by the functional index of the peroneal nerve. The combined therapy group, which received AT-MSC therapy and DMF, exhibited more pronounced effects than the AT-MSC group. The group that received both therapies demonstrated earlier and sustained improvements in motor function throughout the experiment, which surpassed those of the other groups. Evaluation of recovery curves confirmed that the group that received the combined therapy exhibited superior improvement over time.

Regarding the maximum area of paw contact with the platform, minimal or no contact was initially observed, with a subtle improvement from day 39. Notably, the VRR + AT-MSC and VRR + AT-MSC + DMF groups demonstrated enhanced contact, reaching nearly 50% of the baseline value in the combined therapy group. A similar pattern was noted in the force applied by the paw on the platform (maximum intensity), with the group receiving all therapies exhibiting early improvement, approaching baseline values.

Regarding the sequence of paw placement on the walkway (step sequence), it was observed that post-injury, the animals exhibited a deficiency in motor coordination between their paws. The VRR + AT-MSC group showed improved motor coordination towards the experimental endpoint, around day 75, whereas the VRR + AT-MSC + DMF group displayed comparable improvement at an earlier stage, around day 36. The VRR group also showed improvements, starting on day 48, but to a lesser extent (less than 50%) compared to the other groups (above 75%).

A reduction in the base of support between the front paws was observed in the groups that received cell therapy. However, the animals exhibited an increase in the distance between the hindlimbs after injury, which gradually improved over time in the cell therapy groups, with a more significant improvement observed in the VRR + AT-MSC + VRR group.

In conclusion, the combination of therapeutic strategies involving pharmacological treatment, cell therapy, and surgical repair with HFB demonstrated a notably positive impact on gait recovery, persisting until day 84 after the injury. This recovery occurred earlier and exhibited greater efficacy compared to the other treatment groups (Figure 4).

### 3.7. Gene Expression Is Improved by AT-MSC Treatment (Figure 5)

The transcript levels of pro-inflammatory cytokines, specifically *Il6*, were found to be elevated in all experimental groups compared to the unlesioned group. Notably, the VRR + DMF group demonstrated a reduction in *Il6* transcripts when compared to the VRR + AT-MSC + DMF. This indicated that DMF has a mitigating effect on this cytokine mRNA expression.

Conversely, *Tnfa* exhibited a marked increase in the cell therapy groups when compared to all other groups. Therefore, the use of AT-MSCs appears to induce a more pro-inflammatory state, whereas treatment with DMF seems to counteract this state. Gene transcripts for anti-inflammatory cytokines, specifically *Tgfb1*, exhibited increased transcript levels in all groups that received some form of therapy when compared to the unlesioned group. VRR + AT-MSC, VRR + AT-MSC + DMF, and the VRR groups demonstrated higher expression when compared to the VRA group. Notably, the reimplantation with cell therapy group displayed higher *Tgfb1* expression than the VRR and VRR + DMF groups, indicating a more pronounced increase in groups that received cell therapy. Equally, no statistically significant differences in *Il4* cytokine gene expression were observed between the groups.

The analysis of macrophage subtypes based on *Nos2* (M1-type) and *Arg1* (M2-type) markers revealed interesting findings. The *Nos2* gene expression did not show statistical differences among different treatments. On the other hand, *Arg1* expression exhibited notable variability among groups. Both cell therapy groups showed increased expression compared to the unlesioned, VRA, and VRA + DMF groups. The VRR group also demonstrated increased expression compared to the unlesioned, VRA, and VRR + DMF counterparts, with values approaching those of the cell therapy groups. This implies that reimplantation with HFB and treatment with AT-MSCs may positively influence *Arg1* expression, while DMF may not have a similarly positive effect. The balance between *Nos2* and *Arg1*, indicative of the pro-regenerative state, was observed in the group that received the combined therapy.

The analysis of gene expression for trophic factors revealed distinct patterns among the groups. The VRR + DMF group displayed an increase in *Bdnf* levels compared to the unlesioned, VRA, VRR, and VRR + AT-MSC + DMF groups, suggesting that the DMF alone positively modulated *Bdnf* expression. As for *Gdnf*, both groups treated with cells showed elevated expression levels in comparison to all other groups, with exception of the VRR group, implying that treatment with AT-MSCs positively modulated *Gdnf* expression. The VRR + AT-MSC group exhibited an increase in HGF compared to all other groups, except the reimplantation with the cell and DMF groups. The reimplantation-only group also showed an increase in *Hgf* levels compared to the groups without injury and with injury, indicating that treatment with AT-MSCs positively modulated *Hgf* expression. With regard to *Vegfa* only, the VRR + AT-MSC group displayed upregulation in comparison to all other groups, suggesting that treatment with AT-MSCs positively modulated *Vegfa* expression in the spinal cord.

In summary, the results demonstrate that AT-MSCs exerted a favorable influence on the expression of *Gdnf*, *Hgf*, and *Vegfa*, while DMF positively influenced *Bdnf* expression. The combined therapies (reimplantation with AT-MSC and DMF) also showed positive effects on trophic factor expression, indicating a potential synergistic effect.

**Figure 5 biology-13-00953-f005:**
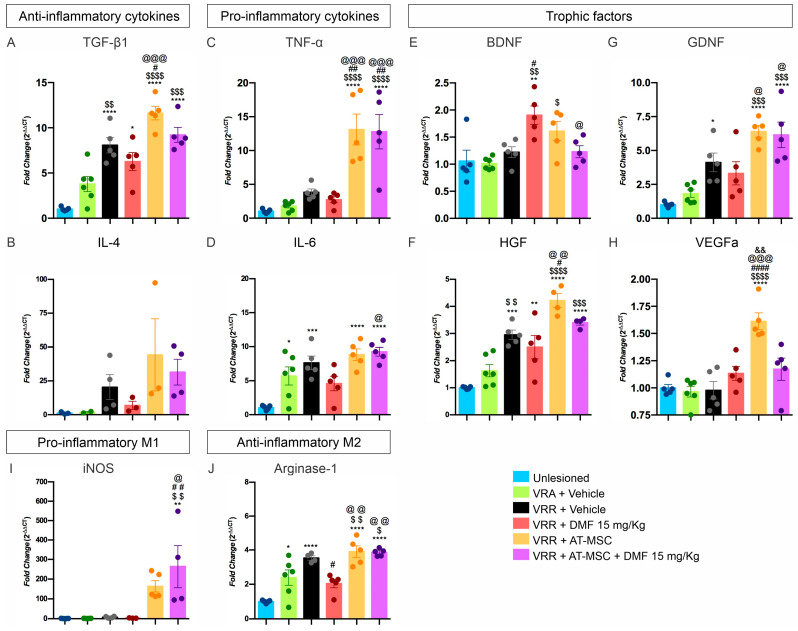
Effects of reimplantation, cell therapy, and DMF treatment on the relative gene expression 7 days after the injury. Anti-inflammatory cytokines (**A**) *Tgfb1* and (**B**) *Il4* and pro-inflammatory cytokines (**C**) *Tnfa* and (**D**) *Il6*. Trophic factors (**E**) *Bdnf*, (**F**) *Hgf*, (**G**) *Gdnf*, and (**H**) *Vegfa*. M1 macrophage marker (**I**) *Nos2* and M2 macrophage marker (**J**) *Arg1*. Mean values ± SEM. * compared to unlesioned, * *p* < 0.05, ** *p* < 0.01, *** *p* < 0.001, and **** *p* < 0.0001; $ compared to VRA, $ *p* < 0.05, $$ *p* < 0.01, $$$ *p* < 0.001, and $$$$ *p* < 0.0001; # compared to VRR, # *p* < 0.05, ## *p* < 0.01, and #### *p* < 0.0001; @ compared to VRR + DMF, @ *p* < 0.05, @@ *p* < 0.01, @@@ *p* < 0.001; & compared to VRR + AT-MSC, && *p* < 0.01.

## 4. Discussion

Recent advancements in cell therapy have raised expectations for new treatment options using MSCs isolated from adult tissues, offering a promising therapeutic strategy for patients with spinal cord injuries. The transplantation of MSCs into the injured spinal cord has been demonstrated to promote neuronal and axonal regeneration, thereby reducing neuronal death within the lesion and facilitating improvements in motor function, as reported in several studies [9,10,34,35,38,47]. This is attributed to their prominent proliferative potential, immunomodulatory abilities, and neuroprotective characteristics.

In this study, we compared the effects of two distinct cell strains, namely BM-MSCs and AT-MSCs, following motor root avulsion. Our observations indicated that both cell types significantly promoted neuronal survival (exceeding 50%) and effectively reduced glial responses, including astrogliosis and microgliosis, thereby regulating the inflammatory response within the tissue. Notably, AT-MSCs demonstrated superior synaptic preservation compared to BM-MSCs.

Prior in vitro studies have shown that AT-MSCs exhibit a superior capacity for mRNA expression of PDGF-b (platelet-derived growth factor) and VEGFA (vascular endothelial growth factor A), whereas BM-MSCs express higher levels of BDNF (brain-derived neurotrophic factor) mRNA [38]. Furthermore, under conditions of hypoxia and oxidative stress, AT-MSCs demonstrate a greater proliferative capacity and metabolic activity than BM-MSCs, which are crucial characteristics in the context of the injured microenvironment [38,48,49].

In the ventral root avulsion model, intense neuronal loss occurs within the first two weeks after injury. This is primarily due to the disruption of the connection between the motoneuron cell body and its axon and the target muscle, resulting in retrograde and anterograde changes, namely chromatolysis and Wallerian degeneration. In the present study, we observed a positive impact on neuronal survival, reaching 44%, following the re-establishment of the connection between the neuronal body and the respective spinal rootles by the ventral root reimplantation with HFB. When reimplantation was combined with MSC cell therapy (both AT-MSC and BM-MSC), an increased survival rate of 57% was observed. This improvement may be mediated by neurotrophic factors and other molecules provided by the MSC therapy as well as by the Schwann cells present in the peripheral nervous system [34,38].

Interestingly, GDNF, identified as the most potent neurotrophic factor to date (up to 75 times more potent than BDNF) [1,50], exhibited a gene transcript fold change greater than 6 in groups treated with MSCs. In contrast, BDNF demonstrated a gene transcript fold change greater than 1.5 but not exceeding 2. Previous studies have indicated that AT-MSCs exhibit a higher capacity to secrete GDNF and a lower capacity to secrete BDNF, whereas BM-MSCs exhibit a greater capacity to secrete BDNF and a lower capacity to secrete GDNF [38]. Additionally, we observed a significant increase in the expression of *Vegfa* gene transcripts in groups that were treated with AT-MSCs. The VEGFa is a factor involved in neovascularization, reinforcing the positive interaction exerted by the cell therapy.

A proposed mechanism for the initiation of apoptosis is through oxidative stress. Increased neurofilament phosphorylation, coupled with higher levels of nitric oxide synthase enzyme activity, is thought to initiate degenerative events [8]. Herein, when DMF was combined with the root repair and cell therapy, we observed a neuronal survival rate of 63%. Since DMF treatment has an antioxidant role [26], and MSCs can contribute to neurotrophic factors, the combination of these therapies resulted in significant neuroprotection.

Synaptic changes, such as retraction and stripping, are also prominent features after proximal axotomy of motoneurons, and have been thoroughly studied by transmission electron microscopy and immunohistochemistry [51]. Synaptic detachment is initially influenced by motoneuron response to injury (autonomously), and later by interaction with glial cells. The extent and pattern of synaptic loss depends on the type of injury and its proximity to the neuronal cell body [52]. Our observations indicate that root reimplantation with HFB preserved the majority of GABAergic inputs but had a reduced capacity to preserve glutamatergic inputs. This is consistent with the concept that a greater loss of excitatory synapses occurs to protect injured neurons from the glutamatergic excitotoxicity [53,54]. Of note, both groups that received cell therapy, with or without DMF association, exhibited a significant reduction in the detachment of excitatory synapses. However, there was an increase in the detachment of inhibitory synapses in comparison with the reimplantation group.

Coupled with motoneuron loss and synaptic pruning after injury, the affected tissue undergoes intense inflammation, leading to reactive gliosis. In this state, glial cells secrete various cytokines and neurotrophic molecules, and take on a phagocytic role that is essential for tissue repair [55]. These reactions can result in both harmful [56] or beneficial [57,58] effects in CNS injury. Our observations show that all therapies, whether used alone or in combination, significantly contributed to the control of tissue inflammation. At 4 weeks, the spinal cord sections showed intense immunostaining for microglia and astrocytes, indicating acute inflammation. Treatment with BM-MSC or AT-MSC led to a significant reduction in inflammation, particularly for the microglial reaction. This effect is likely due to the ability of these cells to produce and secrete various cytokines, thus, modifying the inflammatory milieu by reducing the astrogliosis and microglial reaction [59].

Furthermore, the combined treatment resulted in a higher expression of gene transcripts for the anti-inflammatory cytokines *Il4* and *Tgfb* compared to pro-inflammatory cytokines *Il6* and *Tnfa*. These changes in gene expression were not evident in the reimplantation-only group and, when associated with DMF treatment, they were subtle, highlighting that the anti-inflammatory effects were primarily due to cell therapy. However, in terms of controlling tissue inflammation, the DMF treatment showed a marked reduction compared to the other experimental groups. This reduction can be attributed to DMF’s ability to attenuate cytotoxic and oxidative events and to modulate immune cells, favoring the M2 state of macrophages.

Thus, in the acute phase after injury, motoneurons can enter a state of functional synaptic detachment that coincides with the attempt to regenerate axons. Such a scenario can be reversed by rootlet’s reinnervation and, ultimately, with muscle reinnervation [52]. In this context, DMF and AT-MSC treatments appear to be more supportive, accelerating regeneration, which can significantly contribute to an expressive functional recovery.

Indeed, a subtle improvement in peroneal nerve function was observed herein 39 days after injury, and it was maintained until the twelfth week. However, restoration of the CNS/PNS communication by reimplantation alone is insufficient for long-term functional recovery [14,18]. The cell therapy with AT-MSCs resulted in an earlier improvement, both of strength and motor coordination, which was evident from day 3 after the injury and sustaining until the end of the experiment.

By combining surgical repair, AT-MSCs, and DMF—we achieved axonal regeneration, neuronal survival, control of inflammation, and a reduction in oxidative stress, leading to a significant 70% functional recovery of the peroneal nerve index. However, it is important to emphasize that the present preclinical study requires further validation in larger animal models, including primates, before translation to the clinic. Indeed, we provide proof of concept that the combinatorial treatment approach is effective in a controlled rat model of injury. Thus, the use of female rats for ease of handling and CatWalk analysis could be complemented by further evaluation of male response to the same experimental approach. Although we used inbred Lewis rats, there is also the possibility of individual variation in response to injury. Thus, further follow-up studies are needed to better understand the beneficial effects of pharmacological and stem cell therapy treatment. Finally, the use of AT-MSCs in the clinic needs to be carefully evaluated, as their derivatives, such as exossomes, also show effective results without the need for cell transplantation, which, in turn, may allow for new therapeutic approaches with less risk.

## 5. Conclusions

In the context of spinal cord injury repair and therapy, the present data emphasize the importance of early intervention based on restoration of the CNS/PNS interface. The synergistic application of root reimplantation with HFB, cell therapy with AT-MSCs, and pharmacological treatment with DMF yielded remarkable results. This integrated approach resulted in substantial neuroprotection (63%), a marked reduction in spinal cord inflammation, and preservation of synaptic inputs to spinal motoneurons. Together, these beneficial effects contributed to a significant improvement in functional recovery. Consequently, this work reinforces the need for early combined therapeutic strategies as a viable clinical approach after spinal cord injury.

## Figures and Tables

**Figure 1 biology-13-00953-f001:**
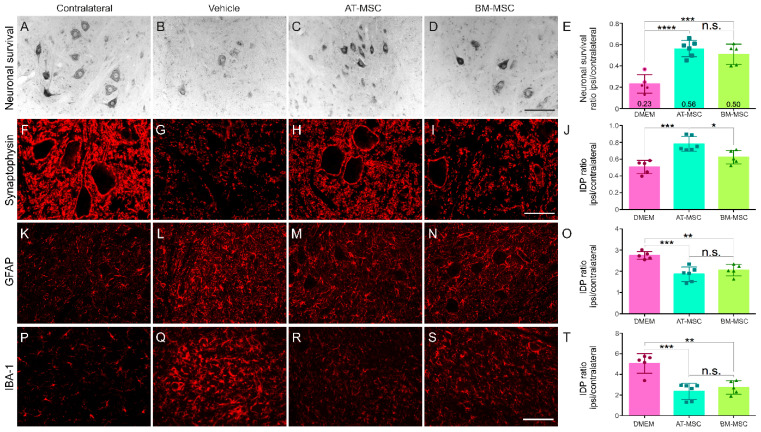
AT-MSCs have a better response than BM-MSCs in neuroprotection and synapse preservation four weeks after motor root avulsion. (**A**–**D**) Histological sections stained with toluidine blue show the Lamina IX of Rexed in the different experimental groups. (**F**–**I**) Immunohistochemical analysis using anti-synaptophysin for synapse visualization, (**K**–**N**) anti-GFAP for astrocyte detection, and (**P**–**S**) anti-IBA-1 for microglia identification in the ventral horn. (**E**) Quantification of the motoneuron survival. Integrated density of pixels quantification showing ipsi/contralateral ratio for the immunomarkers: (**J**) anti-synaptophysin, (**O**) anti-GFAP, and (**T**) anti-IBA1. IDP = Integrated density of pixels. Images captured at 20× magnification for toluidine blue, anti-GFAP, and anti-IBA1 (scale bar = 100 µm) and 40× magnification for anti-synaptophysin (scale bar = 50 µm). N = 5–6. Mean ± SEM. * *p* < 0.05, ** *p* < 0.01, *** *p* < 0.001, and **** *p* < 0.0001, n.s.—no significant.

**Figure 2 biology-13-00953-f002:**
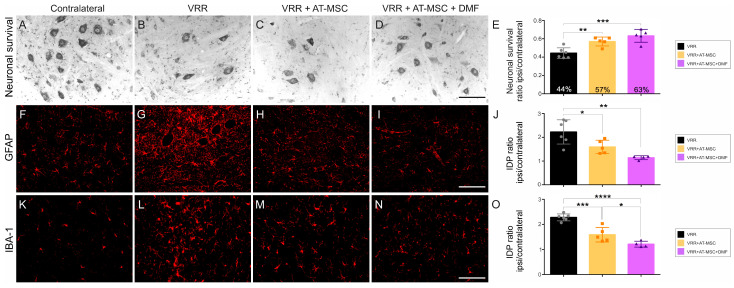
AT-MSCs associated with DMF treatment are neuroprotective and result in decreased glial reactions twelve weeks after motor root avulsion and repair. (**A**–**D**) Cross-sections of the spinal cord ventral horn stained with toluidine blue, showing motoneurons that possess large cell bodies. Note the improved neuronal survival following the combinatory treatment (VRR + AT-MSC + DMF). (**F**–**I**) Immunohistochemical labeling using anti-GFAP antibody to detect astrocyte reactivity. Observe the downregulation of labeling after treatment, and (**K**–**N**) anti-Iba-1 to identify microglia in the anterior column. Observe the significant decrease in microglial reaction following the combined treatments. (**E**) Neuronal survival by the quantification of ipsi/contralateral ratio or surviving motoneurons. (**J**) anti-GFAP and (**O**) anti-Iba1 labeling quantification. IDP = Integrated density of pixels. Images were captured at 20× magnification (scale bar = 100 µm). N = 5–6. Means per group ± SEM. * *p* < 0.05, ** *p* < 0.01, *** *p* < 0.001, and **** *p* < 0.0001.

**Figure 3 biology-13-00953-f003:**
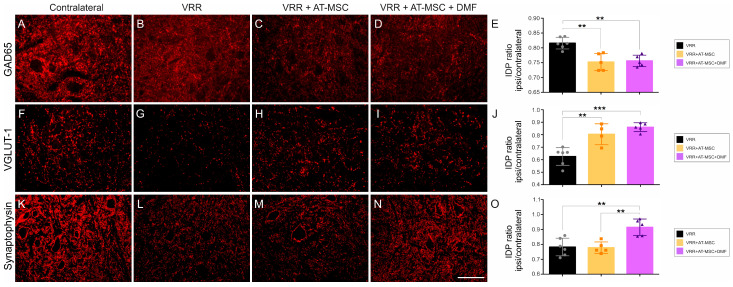
AT-MSCs associated with DMF treatment regulate the balance between excitatory and inhibitory inputs as well as overall pre-synaptic inputs present in the ventral horn twelve weeks after ventral root avulsion. Immunohistochemical analysis using (**A**–**D**) anti-GAD65 for GABAergic inputs detection, depicting a decrease in inhibitory terminals within the neuropil surrounding the motoneurons; (**F**–**I**) anti-VGLUT-1 for glutamatergic inputs detection, showing increase in excitatory synapse nearby the repaired motoneurons; and (**K**–**N**) anti-synaptophysin to detect the overall distribution of synapses around the motoneurons, showing a significantly enhanced preservation of circuitry close by the rescued motoneurons. Quantification of ipsi/contralateral ratio for immunomarkers: (**E**) anti-GAD65, (**J**) anti-VGLUT-1, and (**O**) anti-synaptophysin. IDP = Integrated density of pixels. Scale bar = 100 µm. Data are presented as mean ± SEM. ** *p* < 0.01, *** *p* < 0.001.

**Figure 4 biology-13-00953-f004:**
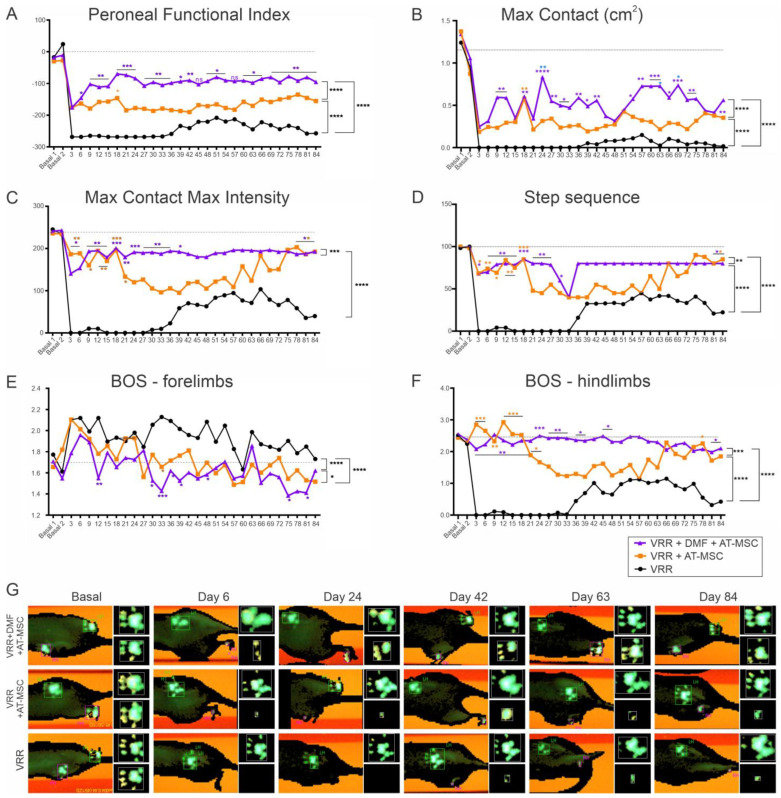
Progressive gait recovery after ventral root reimplantation with HFB, AT-MSC therapy, and DMF treatment. (**A**) Peroneal nerve Functional Index, indicating faster and more robust recovery following the combined therapy. Observe that the ventral root reimplantation alone shows partial recovery over time, which is statistically inferior to cell therapy alone or in combination with DMF. (**B**) Max contact (cm) indicating improvement in the pawprint ipsilateral to the lesion. (**C**) Maximum contact maximum intensity. (**D**) Step sequence. Note the significant recovery in the combined treatment group, which was sustained over time. (**E**) Base of support of front paws. (**F**) Base of support of hind paws. The results indicate a near-normal recovery in the ability to position the limbs during ambulation. (**G**) Representative images of the right (injured) and left (uninjured) paws obtained using the Catwalk System at different times during the experiment, showing the progressive motor recovery after injury and reimplantation of the roots. N = 5–6. * Indicates improvement over other groups. * *p* < 0.05, ** *p* < 0.01, *** *p* < 0.001, **** *p* < 0.0001, n.s.—non significant.

**Table 1 biology-13-00953-t001:** Experiments, survival times, experimental groups, and respective treatments. VRA: ventral root avulsion; VRR: ventral root reimplantation; DMEM: Dulbecco’s Modified Eagle Medium, a control for cell therapy; DMF: dimethyl fumarate 15 mg/kg; AT-MSC: adipose tissue-derived mesenchymal stem cells; and BM-MSC: bone marrow-derived mesenchymal stem cells.

Experiments	Groups		
Experiment I (4 weeks)	DMEM (N = 5)	AT-MSC (N = 6)	BM-MSC (N = 5)
Experiment II (12 weeks)	VRR (N = 6)	VRR + AT-MSC (N = 5)	VRR + DMF + AT-MSC (N = 5)
Experiment III (1 week)	No lesion (N = 5)	VRA (N = 5)	VRR (N = 5)
	VRR + DMF (N = 5)	VRR + AT-MSC (N = 5)	VRR + DMF + AT-MSC (N = 5)

## Data Availability

The datasets generated during the current study are available from the corresponding author.

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
