# Peer review of "Combination of Adult Mesenchymal Stem Cell Therapy and Immunomodulation with Dimethyl Fumarate Following Spinal Cord Ventral Root Repair"

_biology, 2024, doi:10.3390/biology13110953_

Round 1
Reviewer 1 Report
Comments and Suggestions for Authors
Title: Combination of Adult Mesenchymal Stem Cell Therapy and Immunomodulation with Dimethyl Fumarate Following Spinal Cord Ventral Root Repair.
The study investigates the synergistic application of surgical repair, neuroprotection, and immunomodulation to enhance regenerative outcomes following spinal cord ventral root injuries. By combining root reimplantation using heterologous fibrin biopolymer (HFB), dimethyl fumarate (DMF) for neuroprotection and immunomodulation, and adipose tissue-derived mesenchymal stem cells (AT-MSCs), the authors aimed to create a comprehensive therapeutic strategy to support neural repair and improve functional recovery. While the study shows promise in its integrated approach, significant issues with the clarity of the manuscript and the presentation of data make it difficult to fully appreciate the contributions of the findings.
While the study presents valuable insights into combining MSC therapy, DMF, and root reimplantation for spinal cord repair, the manuscript requires significant improvements in clarity, organization, and citation support. Addressing these concerns will enhance the impact of the findings and make the work more accessible to readers.
Below are detailed critiques and suggestions to improve the manuscript:
Major Critiques:
1. Missing Hypothesis:
- The manuscript lacks a clear hypothesis in both the abstract and the introduction sections. While the authors present the objectives of the study, they do not explicitly define the hypothesis. This leaves the reader without a clear understanding of the expected outcomes or rationale behind the study. The hypothesis should clearly state what the researchers aim to demonstrate through their combination therapy of AT-MSCs, DMF, and root reimplantation. Please add this to both the abstract and the introduction for clarity.
2. Missing Citations:
- Several key statements in the introduction require citations to support them. Without proper references, the assertions lack credibility and appear unsupported. Specifically:
- Lines 32-35: “Subsequently, injured motoneurons degenerate as a consequence of internal signaling that activates apoptotic pathways. This may occur if the neuron exhibits low regenerative capacity or an inability to regenerate or maintain connectivity with appropriate networks.”
- This sentence discusses fundamental concepts of neuronal degeneration and apoptosis that must be supported with relevant literature.
- Lines 78-79: “It has been demonstrated that the transplantation of MSC is an efficacious strategy in the acute injury phase, which corresponds to the initiation of apoptotic and inflammatory processes.”
- The claim that MSC transplantation is effective during the acute injury phase also requires appropriate references from recent research to back the statement.
3. Methods:
- Use of 'Eight-Week-Old' and 'Female-Only' Rats:
- The manuscript does not explain the rationale for selecting eight-week-old, female-only rats for the study. It is important to justify why this specific age and sex were chosen. For example, were these choices based on previously established protocols, or was there a specific biological reason that female rats were more suitable for this study? Addressing this would add clarity and scientific validity.
- Experiment II Description:
- The statement in Lines 97-99 — “In Experiment II, the AT-MSC lineage, which had demonstrated superior results in the previous phase, was combined with pharmacological treatment using DMF and lesion repair with HFB” — is unclear. The reference to "superior results" from a previous phase is vague. The authors need to explain what specific results were superior and how these results informed the decision to proceed with this combination therapy. Additionally, the mention of a “previous phase” is confusing—does this refer to a pilot study or a previous experiment within this study? Provide clear details to improve the explanation.
4. Results:
- The clarity and readability of the figures need to be improved significantly. Specifically:
- Figure 2:
- The legend lacks sufficient explanation of the results depicted. It should offer a detailed description of the experimental findings and their significance. Additionally, the text fonts in the figure are too small and difficult to read. Increasing font size and clarity in the figure would improve its readability.
- Figure 3:
- The legend for Figure 3 should be expanded with a more comprehensive explanation of the datas significance. Like Figure 2, the text on both the X and Y axes is too small and needs to be enlarged for better readability.
- Figure 4:
- The X and Y axes in Figure 4 also suffer from small, hard-to-read text. The font size must be increased to ensure clarity and ease of interpretation for the reader. Providing a more detailed legend here as well would enhance the understanding of the depicted results.
5. Limitations of the Study:
- The manuscript does not adequately discuss the limitations of the study, which is essential for any robust scientific analysis. A discussion of potential limitations could include:
- The use of only female rats, which may limit the generalizability of the findings to both sexes.
- The small sample size or the potential variability in outcomes based on biological differences among subjects.
- Any limitations in the applicability of AT-MSC therapy in a clinical setting, including ethical or logistical considerations.
Incorporating these points will provide a more balanced and critical perspective on the study's findings.
Comments on the Quality of English LanguageModerate editing of the English language is required.
Author Response
Reviewer #1
Below are detailed critiques and suggestions to improve the manuscript:
Major Critiques:
- Missing Hypothesis:
- The manuscript lacks a clear hypothesis in both the abstract and the introduction sections. While the authors present the objectives of the study, they do not explicitly define the hypothesis. This leaves the reader without a clear understanding of the expected outcomes or rationale behind the study. The hypothesis should clearly state what the researchers aim to demonstrate through their combination therapy of AT-MSCs, DMF, and root reimplantation. Please add this to both the abstract and the introduction for clarity.
Response: We added a clear hypothesis to the introduction and abstract as follows: “Consequently, we hypothesized that the combination of pharmacological and stem cell therapy approaches can enhance the rescue of avulsed motoneurons as well modulate inflammation, preserve the synaptic circuitry in the motor nucleus (lamina IX of Rexed), and facilitate motor function recovery after ventral root avulsion reimplantation”.
- Missing Citations:
- Several key statements in the introduction require citations to support them. Without proper references, the assertions lack credibility and appear unsupported. Specifically:
- Lines 32-35: “Subsequently, injured motoneurons degenerate as a consequence of internal signaling that activates apoptotic pathways. This may occur if the neuron exhibits low regenerative capacity or an inability to regenerate or maintain connectivity with appropriate networks.”
- This sentence discusses fundamental concepts of neuronal degeneration and apoptosis that must be supported with relevant literature.
Response: Thank you for your input. We added the required citations to support the statements.
- Lines 78-79: “It has been demonstrated that the transplantation of MSC is an efficacious strategy in the acute injury phase, which corresponds to the initiation of apoptotic and inflammatory processes.”
- The claim that MSC transplantation is effective during the acute injury phase also requires appropriate references from recent research to back the statement.
Response: Thank you for your input. We added the required citations to support the statements.
- Methods:
- Use of 'Eight-Week-Old' and 'Female-Only' Rats:
- The manuscript does not explain the rationale for selecting eight-week-old, female-only rats for the study. It is important to justify why this specific age and sex were chosen. For example, were these choices based on previously established protocols, or was there a specific biological reason that female rats were more suitable for this study? Addressing this would add clarity and scientific validity.
Response: The text has been updated as follows: A total of 115 eight-week-old young adult female Lewis rats (LEW/HsdUnib) were used in this study. Such choice was based on previous literature and in the fact that young adult rats are more suitable to long-term spinal cord injury and repair approaches [14, 18, 39, 43, 44]. Female rats were used due to the long-term limited size and weight gain, which facilitate handling and functional tests such as the walking track test.
- Experiment II Description:
- The statement in Lines 97-99 — “In Experiment II, the AT-MSC lineage, which had demonstrated superior results in the previous phase, was combined with pharmacological treatment using DMF and lesion repair with HFB” — is unclear. The reference to "superior results" from a previous phase is vague. The authors need to explain what specific results were superior and how these results informed the decision to proceed with this combination therapy. Additionally, the mention of a “previous phase” is confusing—does this refer to a pilot study or a previous experiment within this study? Provide clear details to improve the explanation.
Response: We have updated the phrase as follows: In Experiment II, the AT-MSC lineage, which had demonstrated superior results in terms of synaptic preservation, was combined with pharmacological treatment using DMF and lesion repair with HFB. Also, we rephrased the “phase” concept as follows: Table 1 shows the experiments carried out as well as the respective experimental groups.
- Results:
- The clarity and readability of the figures need to be improved significantly. Specifically:
- Figure 2:
- The legend lacks sufficient explanation of the results depicted. It should offer a detailed description of the experimental findings and their significance. Additionally, the text fonts in the figure are too small and difficult to read. Increasing font size and clarity in the figure would improve its readability.
Response: Figure 2 has been updated to facilitate visualization of the lettering. Also, the legend has been improved. The legend has been updated as follows:
Figure 2. AT-MSCs associated with DMF treatment is neuroprotective and results in decreased glial reactions twelve weeks after motor root avulsion and repair. (A-D) Cross sections of the spinal cord ventral horn stained with toluidine blue, showing motoneurons that possess large cell bodies. Note the improved neuronal survival following the combinatory treatment (VRR+AT-MSC+DMF). (F-I) Immunohistochemical labeling using anti-GFAP antibody to detect astrocyte reactivity. Observe the downregulation of labeling after treatment, and (K-N) anti-Iba-1 to identify microglia in the anterior column. Observe the significant decrease of microglial reaction following the combined treatments (E) Neuronal survival by the quantification of ipsi/contralateral ratio or surviving motoneurons. (J) anti-GFAP, and (O) anti-Iba1 labeling quantification. IDP = Integrated density of pixels. Images were captured at 20X magnification (scale bar = 100 µm). N = 5-6. Means per group ± SEM. * p<0.05, ** p<0.01, ***p<0.001, and ****p<0.0001.
- Figure 3:
- The legend for Figure 3 should be expanded with a more comprehensive explanation of the data significance. Like Figure 2, the text on both the X and Y axes is too small and needs to be enlarged for better readability.
Response: Figure 3 has been updated to facilitate visualization of the lettering. Also, the legend has been improved. The legend has been updated as follows:
Figure 3. AT-MSCs associated with DMF treatment regulate the balance between excitatory and inhibitory inputs as well as overall pre-synaptic inputs present in the ventral horn, twelve weeks after ventral root avulsion. Immunohistochemical analysis using (A-D) anti-GAD65 for GABAergic inputs detection, depicting a decrease of inhibitory terminals within the neuropil surrounding the motoneurons (F-I) anti-VGLUT-1 for glutamatergic inputs detection, showing increase of excitatory synapse nearby the repaired motoneurons and, (K-N) anti-synaptophysin to detect the overall dis-tribution of synapses around the motoneurons, showing a significantly enhanced preservation of circuitry close by the rescued motoneurons. Quantification of ipsi/contralateral ratio for im-munomarkers: (E) anti-GAD65, (J) anti-VGLUT-1, (O) anti-synaptophysin. IDP = Integrated density of pixels. Scale bar = 100 µm. Data is presented as mean ± SEM. * p<0.05, ** p<0.01, ***p<0.001, and ****p<0.0001.
- Figure 4:
- The X and Y axes in Figure 4 also suffer from small, hard-to-read text. The font size must be increased to ensure clarity and ease of interpretation for the reader. Providing a more detailed legend here as well would enhance the understanding of the depicted results.
Response: Figure 4 has been updated to facilitate visualization of the lettering. Also, the legend has been improved. The legend has been updated as follows:
Figure 4. Progressive gait recovery after ventral root reimplantation with HFB, AT-MSC therapy and DMF treatment. (A) Peroneal nerve Functional Index, indicating faster and more robust recovery following the combined therapy. Observe that the ventral root reimplantation alone shows partial recovery over time, statistically inferior to cell therapy alone or in combination with DMF. (B) Max contact (cm) indicating improvement in the pawprint ipsilateral to the lesion. (C) Maximum contact maximum intensity. (D) Step sequence. Note the significant recovery in the combined treatment group, which was sustained overtime. (E) Base of support of front paws, and (F) Base of support of hind paws. The results indicate a near-normal recovery in the ability to position the limbs during ambulation. (G) Representative images of the right (injured) and left (uninjured) paws obtained using the Catwalk System at different times during the experiment, showing the progressive motor recovery after injury and reimplantation of the roots. N = 5-6. *Indicates improvement over other groups.
- Limitations of the Study:
- The manuscript does not adequately discuss the limitations of the study, which is essential for any robust scientific analysis. A discussion of potential limitations could include:
- The use of only female rats, which may limit the generalizability of the findings to both sexes.
- The small sample size or the potential variability in outcomes based on biological differences among subjects.
- Any limitations in the applicability of AT-MSC therapy in a clinical setting, including ethical or logistical considerations.
Response: We added a paragraph regarding the limitations of the study as suggested.
However, it is important to emphasize that the present preclinical study requires further validation in larger animal models, including primates, before translation to the clinic. Indeed, we provide proof of concept that the combinatorial treatment approach is effective in a controlled rat model of injury. Thus, the use of female rats for ease of handling and CatWalk analysis could be complemented by further evaluation of male response to the same experimental approach. Although we used inbred Lewis rats, there is also the possibility of individual variation in response to injury. Thus, further follow-up studies are needed to better understand the beneficial effects of pharmacological and stem cell therapy treatment. Finally, the use of AT-MSCs in the clinic needs to be carefully evaluated, as their derivatives, such as exossomes, also show effective results without the need for cell transplantation, which in turn may allow new therapeutic approaches with less risk.
Reviewer 2 Report
Comments and Suggestions for Authors
This study aimed to explore spinal cord injury repair and therapy, with a focus on the critical importance of early intervention to restore the CNS/PNS interface. The integrated approach, combining root reimplantation with HFB, cell therapy using AT-MSCs, and pharmacological treatment with DMF, demonstrated noteworthy outcomes. Overall, this paper is thoroughly written with a well-structured and rigorous framework.
Specific comments:
1. Please provide the passage number of MSCs in the “Results” section (3.1. Characterization of MSC cultures).
2. Please enhance the clarity of the images in Figure 3 (A-D) for anti-GAD65 used in detecting GABAergic inputs, as they appear blurry and lack resolution.
Comments on the Quality of English Language
none
Author Response
This study aimed to explore spinal cord injury repair and therapy, with a focus on the critical importance of early intervention to restore the CNS/PNS interface. The integrated approach, combining root reimplantation with HFB, cell therapy using AT-MSCs, and pharmacological treatment with DMF, demonstrated noteworthy outcomes. Overall, this paper is thoroughly written with a well-structured and rigorous framework.
Specific comments:
- Please provide the passage number of MSCs in the “Results” section (3.1. Characterization of MSC cultures).
Response: We highlighted the information: At the fourth passage, cells were collected, resuspended in DMEM, and applied to the spinal cord at the time of motor root injury in the section 2.2. MSC Isolation and Culture.
- Please enhance the clarity of the images in Figure 3 (A-D) for anti-GAD65 used in detecting GABAergic inputs, as they appear blurry and lack resolution.
Response: We provided a redesigned Figure 3 so that the images are clearer and the lettering is larger, facilitating visualization.
Reviewer 3 Report
Comments and Suggestions for Authors
This paper aims to demonstrate the therapeutic potential for spinal cord ventral root repair by combining dimethyl fumarate (DMF) with mesenchymal stem cells from bone marrow and adipose tissue. Wholly, the data was clear and likely supported the author‘s conclusion at one look. However, these data did not achieve the credible information, because of the lack of main proof.
1. The biggest problem of this paper is a fall through the MSC (BM-MSCs and AT-MSCs) transplantation experiment, because there were no proof of the engraftment of these stem cells, even both cells have been harvested from eGFP-Tg female Lewis rats. Thus, this study actually showed the effects of VRR and DMF. Why the authors did not show the evidence of transplanted stem cells. No evidence of the engraftment and migration for the tissue is likely the problem in common in the same kind of BM- and AT-MSCs transplantation studies through references. Thus, it is impossible to bereave the data without evidence of in-vivo engraftment in the tissues.
2. All photographs are higher magnifications, and there were not clear the relationship to the tissue damage. This is much important if there is no evidence of the trace of the transplanted stem cells, to avoid a doubt of convenience selection.
3. For the reasons mentioned above, I am unable to believe the data from this study.
Author Response
This paper aims to demonstrate the therapeutic potential for spinal cord ventral root repair by combining dimethyl fumarate (DMF) with mesenchymal stem cells from bone marrow and adipose tissue. Wholly, the data was clear and likely supported the author‘s conclusion at one look. However, these data did not achieve the credible information, because of the lack of main proof.
- The biggest problem of this paper is a fall through the MSC (BM-MSCs and AT-MSCs) transplantation experiment, because there were no proof of the engraftment of these stem cells, even both cells have been harvested from eGFP-Tg female Lewis rats. Thus, this study actually showed the effects of VRR and DMF. Why the authors did not show the evidence of transplanted stem cells. No evidence of the engraftment and migration for the tissue is likely the problem in common in the same kind of BM- and AT-MSCs transplantation studies through references. Thus, it is impossible to bereave the data without evidence of in-vivo engraftment in the tissues.
Response: The cells were engrafted together with a fibrin scaffold during the surgery. We have already shown in different publications that they stay in the site of transplantation. Also, the results shown in Figure 1 reinforce the immunomodulation and synaptic preservation, that do not happen in the vehicle counterpart. Thus, the cells are performing their effects as expected.
- All photographs are higher magnifications, and there were not clear the relationship to the tissue damage. This is much important if there is no evidence of the trace of the transplanted stem cells, to avoid a doubt of convenience selection.
Response: We have recently published the surgical procedures in the following paper: https://doi.org/10.1590/1678-9199-JVATITD-2019-0093 (Figure 1), reproduced below. We have indicated that in the revised version of the Material and Methods.
Round 2
Reviewer 1 Report
Comments and Suggestions for Authors
I am delighted to commend the authors for their meticulous attention to the reviewers feedback, which is evident in the revised manuscript. They have addressed all the points raised and integrated the suggested changes, leading to substantial enhancements. These revisions have brought about a notable improvement in the manuscripts clarity and organization, making the content more readily comprehensible. The authors conscientious efforts have truly enriched the overall quality of the article, making it a more engaging and accessible piece of work.
Author Response
Response: We are grateful for the reviewer's insights and appreciate the efforts to improve the manuscript.
Reviewer 3 Report
Comments and Suggestions for Authors
Obviously, I had read the author's previous paper. However, that one was also problematic. The request I have is straightforward. Why did the authors not show co-staining such as GFP+GFAP (N200, etc.)? In the previous paper, injected cells were detected after 12 weeks. What type of cells were they? This is also the case present study. It is obvious that many papers published about MSC (in particular BM-MSC) could not make this point. This is sound in terms of biology, then, everybody admits.
Author Response
Response: In agreement with the reviewer's suggestion, we performed immunolabeling experiments to track the engrafted cells using anti-CD90 antibody in combination with GFAP, Iba-1 and neurofilament. The results are shown in new Supplementary Figure S3. We show the presence of engrafted AT-MSCs 4 weeks after ventral root coaptation within the roots as well as in the environment of the motor neurons. Thus, we hope that the question of MSC survival over time has been sufficiently addressed, demonstrating the benefits of such a treatment strategy. We thank the reviewer for raising this important point.